# Incidence, Clinical Profile, and Cardiac Manifestations of MIS-C in Children in Kuwait

**DOI:** 10.3390/diagnostics15192545

**Published:** 2025-10-09

**Authors:** Ozayr Mahomed, Adnan Alhadlaq, Khaled Alsaeid, Aisha Alsaqabi, Fouzeyah Othman, Saja Al-Shammari, Sarah Al-Yaqoub, Abdullah Al-Daihani, Abdulla Alfraij, Khalid Alafasy, Mafaza Al-Qallaf, Mariam Al-Hajeri, Nora Al-Mutairi, Alaa Alenezi, Shaimaa Mohammed, Adnan Al-Sarraf, Dalia Al-Abdulrazzaq, Hessa Al-Kandari

**Affiliations:** 1Department of Population Health, Dasman Diabetes Institute, P.O. Box 1180, Dasman 15462, Kuwait; mahomedo@ukzn.ac.za (O.M.); aisha.alsaqabi@dasmaninstitute.org (A.A.); fao04@mail.aub.edu (F.O.); dalia.alabdulrazzaq@dasmaninstitute.org (D.A.-A.); 2Discipline of Public Health Medicine, University of KwaZulu Natal, Durban 4041, South Africa; 3Department of Pediatrics, Chest Disease Hospital, Kuwait City 70001, Kuwait; adnan.alhadlaq@gmail.com; 4Department of Pediatrics, Farwaniya Hospital, P.O. Box 13373, Farwaniya 81004, Kuwait; sajaalshemmari@gmail.com (S.A.-S.); alyaqoubs@gmail.com (S.A.-Y.); abdullah_aldaihani@hotmail.com (A.A.-D.); alfraij.abdulla@gmail.com (A.A.); kmalafasy@gmail.com (K.A.); 5Department of Pediatrics, Faculty of Medicine, Kuwait University, P.O. Box 24923, Safat 13110, Kuwait; khaled.alsaeid@ku.edu.kw; 6Pediatric Intensive Care Unit, Department of Pediatrics, General Ahmadi Hospital, Kuwait Oil Company (KOC), Al-Ahmadi P.O. Box 9758, Kuwait; 7Department of Pediatrics, Amiri Hospital, P.O. Box 4077, Safat 13041, Kuwait; mafaza.alqallaf@gmail.com; 8Department of Pediatrics, Adan Hospital, P.O. Box 46969, Fahaheel 64020, Kuwait; maryamhajeri@gmail.com; 9Department of Pediatrics, Sabah Hospital, Kuwait City 70001, Kuwait; nooordr@yahoo.com; 10Department of Pediatrics, Jahra Hospital, Al-Jahra 00042, Kuwait; alaahalenezi@gmail.com (A.A.); shoooeldbsy@gmail.com (S.M.); 11Department of Pediatrics, Mubarak Al-Kabeer Hospital, P.O. Box 43787, Hawalli 32052, Kuwait; alsarrafadnan@gmail.com; 12Ministry of Health, P.O. Box 5, Safat 13001, Kuwait

**Keywords:** multisystem inflammatory syndrome in children, Severe Acute Respiratory Syndrome Coronavirus 2 (SARS-CoV-2), cardiac manifestations, Kuwait

## Abstract

**Background/Objectives**: Multisystem inflammatory syndrome in children (MIS-C), a rare but serious post-acute hyperinflammatory condition that occurs in children 2–6 weeks after Severe Acute Respiratory Syndrome Coronavirus 2 (SARS-CoV-2) infection or exposure, varies between countries. Despite its serious nature, most children recover without any sequelae. The most frequently reported long-term sequelae are coronary artery aneurysms. This study aimed to describe the epidemiological profile, clinical characteristics (including cardiac manifestations), treatment, and outcomes of multisystem inflammatory syndrome in children (MIS-C) under 14 years of age with SARS-CoV-2 between February 2020 and November 2021 in Kuwait. **Methods**: Data on sociodemographic factors, co-morbidities, presenting signs and symptoms, as well as laboratory and echocardiography findings were retrieved from the Pediatric COVID registry (PCR-Q8 registry). **Results**: Of the one hundred and two patients with a provisional diagnosis of MIS-C, eighty-three patients fulfilled the WHO criteria of MIS-C. Thirty-nine of the MIS-C patients were admitted to the intensive care unit, and only one child died due to cardiogenic shock. Sixteen patients from the pediatric MIS-C cohort were diagnosed with cardiac abnormalities. Sixteen patients from the pediatric MIS-C cohort were diagnosed with cardiac abnormalities. Most (63% (10/16)) of the patients had coronary abnormalities, nine patients (56%) had myocardial dysfunction, and six patients (38%) had dual pathologies. Pericarditis occurred in three patients only, whilst six patients (38%) had dual pathologies. Pericarditis occurred in three patients only. **Conclusions**: MIS-C appears to affect younger children in Kuwait than in other countries; however, the clinical pattern is consistent with other countries. Further studies of an analytical nature are recommended to identify the risk factors associated with MIS-C and its cardiac sequalae to allow for proactive risk reduction.

## 1. Introduction

Multisystem inflammatory syndrome (MIS-C), a post-acute hyperinflammatory condition that occurs in children 2–6 weeks after SARS-CoV-2 infection or exposure associated with Severe Acute Respiratory Syndrome Coronavirus 2 (SARS-CoV-2), is a rare but serious condition [1]. The condition was first reported in 2020 in the United Kingdom, followed by many other countries, including Canada, South Africa, and the United States of America [2]. Most of the cases occurred during the Omicron variant phase of the SARS-CoV-2 pandemic [3]. The incidence estimates of MIS-C vary between countries and are influenced by the genetic predisposition of the population as well as the availability of active screening programs [4]. Most cases of MIS-C develop 2–6 weeks after SARS-CoV-2 infection [5], with the majority occurring in previously healthy children [6]. MIS-C has a good prognosis, with 70–90% of children recovering without sequelae even if the initial presentation is severe [7]. A minority (2.41%) of patients presenting with MIS-C demise [8]. Coronary artery aneurysms have been reported most frequently in the literature as the long-term sequela of MIS-C [9].

The exact pathogenic mechanism causing MIS-C is unknown, but it has been postulated that MIS-C may be due to the direct effect of the SARS-CoV-2 spike protein structure on immune activation [10] or a postinfectious immune activation that causes the release of excessive amounts of proinflammatory cytokines, mainly tumor necrosis factor (TNF) alpha, IL-1, and IL6. This is accompanied by a decrease in lymphocytes, such as NK cells, CD4 T lymphocytes, and B lymphocytes [11], contributing to a hyperinflammatory state that causes endothelial dysfunction and multiorgan damage [12]. Risk factors associated with MIS-C in children include, among others, male sex, an age of 5–11 years, asthma, obesity, and other conditions associated with a shorter life expectancy for which no curative treatment is available [13,14].

The first case of pediatric SARS-CoV-2 in Kuwait was identified at the end of February 2020, and the pandemic spread rapidly in the country [15]. The profile of the children diagnosed with SARS-CoV-2 infection in Kuwait (male sex, children between 5 and 11 years of age, obesity, asthma, and presence of co-morbid diseases) [13] was highly suggestive of an increased risk of MIS-C. Given the unique population profile of Kuwait, there is a need to identify the risk factors for MIS-C to identify vulnerable children and initiate targeted population and clinical interventions. This study aimed to describe the incidence and clinical profile of MIS-C and its associated cardiac abnormalities amongst children in Kuwait between 2020 and 2021.

## 2. Materials and Methods

### 2.1. Study Design and Study Population

A retrospective cohort study was conducted. The study population included all children under 14 years of age with SARS-CoV-2 who were hospitalized in Kuwait and had their data recorded in the Pediatric COVID-19 registry between February 2020 and November 2021.

### 2.2. Data Sources

The Pediatric COVID-19 Task Force at the Ministry of Health (MOH) initiated the creation and implementation of the SARS-CoV-2 registry for children under 14 years of age (PCR-Q8 registry) [16]. Primary data sources for the registry included all facilities where SARS-CoV-2 testing was conducted: the nation’s six government hospitals (Farwaniya, Al-Amiri, Al-Sabah, Mubarak Al-Kabeer, Al-Adan, and Al-Jahra) and one COVID-19 referral center (Jaber Al-Ahmad COVID-19 Referral Centre). Additional secondary data sources were gathered from primary care centers, quarantine facilities, the dispatch call center, and public health laboratories (including results from drive-thru swabs) [16].

Kuwait’s Pediatric COVID-19 Task Force adopted the WHO preliminary MIS-C definition (15 May 2020) as the operational standard for case confirmation and national surveillance (Table 1). The RCPCH PIMS-TS (UK) describes a similar pediatric hyper-inflammatory syndrome but does not mandate laboratory/epidemiologic linkage to SARS-CoV-2; the CDC MIS-C definition is closely aligned with that of the WHO but differs modestly in age and fever thresholds. In our registry, children who met PIMS-TS features without a WHO-compatible SARS-CoV-2 link were not classified as MIS-C (Table 1). The definition was derived from clinical and laboratory features observed in children to identify suspected or confirmed cases, both for treatment purposes and for provisional reporting and surveillance [17].

### 2.3. Classification of MIS-C

MIS-C severity is categorized as mild, moderate, and severe, and disease severity classification is determined by the Vasoactive Inotropic Score (VIS), the degree of respiratory support, and evidence of organ injury [18]. Mild cases have no vasoactive requirement, minimal respiratory support, and minimal signs of organ injury. Moderate cases have a VIS ≤ 10, significant supplemental oxygen requirement, and mild or isolated organ injury. Severe cases have a VIS > 10, non-invasive or invasive ventilatory support, and moderate or severe organ injury, including moderate to severe ventricular dysfunction [19].

### 2.4. Data Collection

Data were retrieved from the PCR-Q8 registry. The variables collected included the demographic profile of the patient (age, gender, and nationality), the presence or absence of clinical signs and symptoms at presentation, co-morbidities, diagnosis and severity of MIS-C (mild, moderate, or severe), treatment provided, diagnostic radiology findings (chest X ray on day 0 and day 7), laboratory investigations (white blood cells, neutrophils, lymphocytes, C-reactive protein, troponin T, ferritin, D-dimer, platelet, hemoglobin, interleukin-6), need for pediatric intensive care unit (PICU) admission, and the outcome of the condition. In addition, findings from the electrocardiogram (ECG) and echocardiographic examinations were retrieved. Cardiac findings were classified as follows: myocardial dysfunction (LVEF < 55% or FS < 28%); pericarditis (effusion/inflammation); valvulitis (new ≥ mild regurgitation/leaflet thickening); coronary artery abnormalities per AHA z-scores—dilatation/ectasia (z 2.0–<2.5), small aneurysm (z 2.5–<5.0), medium (z 5.0–<10.0), and large/giant (z ≥10 or ≥8 mm); and pulmonary hypertension (estimated RVSP > 40 mmHg or accepted surrogates). Where z-scores were absent, absolute diameters/qualitative impressions were recorded.

### 2.5. Quality and Reliability of Diagnosis

Although the case definition was used to make a preliminary diagnosis, the final diagnosis of MIS-C was made by a team of clinicians that included a pediatric infectious disease specialist, pediatric rheumatologist, pediatric cardiologist, and general pediatricians. Once the diagnosis was confirmed, the patient was labeled in the Pediatric COVID-19 registry as a case of MIS-C. To improve the validity of the findings, a pediatric cardiologist independently reviewed echocardiogram scans of 15 patients blindly. There was no divergence from the initial records of echocardiogram findings.

### 2.6. Case Ascertainment and Adjudication

Children flagged as provisional MIS-C in the registry underwent multidisciplinary case review (pediatric infectious diseases, rheumatology, cardiology, and general pediatrics) against the WHO preliminary MIS-C definition. Confirmation required fulfillment of clinical, laboratory, and epidemiologic criteria; cases failing to meet full criteria or with a more likely alternative diagnosis were not classified as MIS-C.

### 2.7. Data Analysis

Data were entered into Microsoft Excel and thereafter exported. Data were analyzed in Stata 18 (Stata Corp LLC, College Station, TX, USA). Frequencies (counts and proportions) were determined for demographic, severity of MIS-C, and clinical characteristics. The incidence proportion was calculated based on the number of MIS-C cases diagnosed in relation to the total number of children with SARS-CoV-2 recorded in the PCR-Q8 registry, as well as the number of MIS-C cases diagnosed in relation to the number of children with SARS-CoV-2 admitted at hospital. Cardiac abnormalities were assessed based on echocardiogram results. All children meeting MIS-C criteria received echocardiography and ECG by protocol. Analyses were conducted on available cases, and no single or multiple imputation was performed. For each variable, the analysis denominator reflects the number of non-missing observations and is reported in the text/tables as *n/N*. Variables captured by protocol for all MIS-C cases (case status, MIS-C severity, PICU admission, echocardiography, and ECG) were complete. No values were substituted or carried forward.

Group comparisons used Pearson’s χ^2^ (or Fisher’s exact for sparse/2 × 2 tables) for associations of the cardiac outcome with sex, age group, nationality, and MIS-C severity. Distributions of continuous biomarkers (CRP, ferritin, D-dimer, troponin) by cardiac outcome were compared using the Mann–Whitney U test. LVEF was compared with cardiac outcome using Mann–Whitney U, and its association with troponin was summarised with Spearman’s rank correlation (ρ). All tests were two-sided (α = 0.05)**;** exact *p*-values are reported. Exploratory cardiac associations used Mann–Whitney U to compare biomarker distributions by echocardiographic status and Spearman’s ρ for correlations between LVEF and biomarkers (two-sided α = 0.05; available-case denominators; no imputation or multiplicity adjustment).

### 2.8. Ethics

The study was conducted according to the guidelines of the Declaration of Helsinki and was approved by the Ministry of Health in Kuwait (1597/2020) on 30 December 2020. As the study involved secondary data analysis with fully anonymized data and no direct patient contact, informed consent was not required.

## 3. Results

### 3.1. Incidence of MIS-C Amongst Children with SARS-CoV-2

Between February 2020 and November 2021, data for 24,637 children diagnosed with pediatric SARS-CoV-2 were documented in the PCR-Q8 registry. Of these, 1599 patients with SARS-CoV-2 were admitted to hospitals. One hundred and two patients (6%) had a provisional diagnosis of MIS-C. Of the 102 patients, 83 had a final diagnosis of MIS-C; most of the admitted children (59%) had mild MIS-C (Figure 1).

The incidence of MIS-C translates to 3.32 cases per 1000 registered cases (95% CI ≈ 2.7–4.2 per 1000), or 5 cases of MIS-C per 100 hospital admissions for SARS-CoV-2. There were two peaks in hospital admissions for MIS-C: June 2020 and August 2021 (Figure 2). Forty-two percent of the children with MIS-C were admitted during the first wave of SARS-CoV-2 (‘WH-Human 1’ coronavirus, referred to as ‘2019-nCoV’) [20] between February 2020 and November 2020. There was a decline in incidence during the second wave (Alpha variants (B.1.1.7)) and third wave (Beta variant phase (B 1.1351)). There was a subsequent rise in incidence, as 39% of cases occurred in the fourth wave (Delta variant phase (B.1.617.2 strain)).

### 3.2. Demographic and Clinical Characteristics of the Children with MIS-C

The mean age of children with MIS-C was 6.07 years (SD: 3.44), with 59% (50) children being under 6 years of age. There were more males than females (Table 2 (a)). Fifty-two percent of the children admitted with MIS-C were non-Kuwaiti residents of foreign origin (mainly Egyptian, Indian, and Syrian). Eight patients had pre-existing medical conditions such as asthma (three), chronic kidney disease (three), acquired/congenital heart disease (two), and hypertension (one). In total, 30 percent (25) of the children with MIS-C displayed evidence of a recent SARS-CoV-2 infection, with a positive PCR COVID-19 IgM antibody, and 24% (24) showed positive IgG antibodies. Thirty-four percent of children with MIS-C (29/84) had positive PCR tests on Nasopharyngeal swabs for COVID-19. There were more non-Kuwaiti males (28 (34%)) diagnosed with MIS-C compared to Kuwaiti males (*n* = 15, 18%), while more Kuwaiti females (*n* = 24, 29%) were diagnosed with MIS-C compared to non-Kuwaiti females (*n* = 15, 18%). More males (8) than females (4) had severe MIS-C, whereas more females (14) than males had moderate MIS-C. Patients with MIS-C had high ferritin, troponin -t, and D-dimer counts (Table 2 (b)).

All patients presented with a history of fever. The mean duration of fever to admission was 5.5 days (SD: 3.91). Gastrointestinal symptoms such as vomiting (71%), nausea (65%), abdominal pain (51%), and anorexia (36%) were the most common presenting symptoms for patients with MIS-C. Sixty-nine percent of patients presented with bilateral conjunctivitis and skin rashes, respectively. Respiratory symptoms such as cough, sore throat, rhinorrhoea, and shortness of breath were experienced by 27%, 25%, 16%, and 12% of the patients, respectively. A quarter (25%) of the patients presented with irritability, 19% had muscle aches, 14% had headaches, and 13% presented with an inability to walk. Respiratory signs such as pleural effusion (13%) and viral pneumonia (10%) were amongst the most common clinical signs detected in patients with MIS-C (Figure 3).

### 3.3. Echocardiogram Abnormalities

Sixteen patients from the pediatric MIS-C cohort were diagnosed with cardiac abnormalities based on echocardiogram findings. Of the sixteen patients with MIS-C-related cardiac abnormalities, five children had severe/critical COVID disease (SpO_2_ < 94% on room air, PaO_2_/FiO_2_ ratio < 300, a respiratory rate >30 breaths/min, or lung infiltrates >50%), six children had moderate COVID disease (lower respiratory disease during clinical assessment or imaging, with an SpO_2_ ≥ 94% on room air) and five children had mild SARS-CoV-2 infection (no shortness of breath, dyspnea on exertion, or abnormal imaging) [21].

Most (63% (10/16)) of the patients had coronary abnormalities, nine patients (56%) had myocardial dysfunction, and six patients (38%) had dual pathologies. Pericarditis occurred in three patients only (Figure 4). The left ventricular ejection fraction (LVEF) was available for 50% of the patients. The mean LVEF for the sample was 61.25 (SD 9.44); however, there was a significant difference (*p* < 0.05) between the mean LVEF of patients with no cardiac abnormalities (63.44) in comparison to patients with cardiac abnormalities detected (56.92).

### 3.4. Demographic and Laboratory Profile of Patients with Cardiac Abnormalities

Across the cohort, the mean biomarker levels did not differ significantly by the presence of any echocardiographic/cardiovascular abnormality: ferritin (*p* = 0.854), CRP (*p* = 0.120), troponin T (*p* = 0.136), and D-dimer (*p* = 0.147). Cardiac abnormality was not associated with sex (χ^2^ = 0.58, *p* = 0.45) or age group (χ^2^ = 4.5, *p* = 0.21). There was a modest association with nationality, with a higher proportion of abnormalities among non-Kuwaiti children (χ^2^ = 4.06, *p* = 0.04) (Table 3).

In a cardiac-focused severity analysis (available cases, *N* = 82), the proportion with any echocardiographic abnormality was 21.7% in mild, 22.7% in moderate, and 8.3% in severe disease. Overall, disease severity was not statistically associated with cardiac abnormality (χ^2^ = 1.73, df = 3, *p* = 0.6303) (Table 3).

Compared with children without echocardiographic abnormalities, those with any abnormality showed a non-statistically significant higher troponin concentration (median 12.00 [IQR 7.37–21.40] vs. 4.96 [0.45–22.84] ng/L), (Mann–Whitney U *p* = 0.2161). D-dimer, ferritin, and CRP were similar between groups (Table 4). LVEF was lower among children with any echocardiographic abnormality (median 56.0% [50.5–65.0]), although the difference vs. those without abnormality was not significant (Mann–Whitney U *p* = 0.1003). In correlation analyses, troponin was inversely related to LVEF (Spearman ρ = −0.199, *p* = 0.3208), while D-dimer (ρ = 0.183, *p* = 0.3245) and ferritin (ρ = 0.111, *p* = 0.5732) showed weak positive, non-significant correlations, and CRP trended inversely (ρ = −0.271, *p* = 0.1045). Overall, there were no biomarker–echo associations in this cohort, with estimates limited by sample size and missing values.

### 3.5. Treatment and Outcomes

Thirty-nine (47%) of the MIS-C patients required intensive care (ICU). The mean duration of stay in the ICU was 5.7 days (SD 3.95). Eight (9%) of the children with MIS-C required high-flow nasal cannula, and five (6%) patients required mechanical ventilation. Seventy-one patients with MIS-C received intravenous immunoglobulins, with fifty-eight patients also receiving corticosteroids. Fifteen patients required ionotropic support.

Fifty-four patients (68%) recovered fully. Twenty-seven patients had residual cardiac sequelae from the MIS-C. Eighteen patients had residual left ventricular dysfunction (LVEF < 55%) post MIS-C, with an additional two patients having aneurysms (one classified as small (z-score: 2.5–5) and the other medium size (z-score 5–10)) at the three months follow up. Among those with follow-up echocardiography at approximately 3 months (*n* = 18), 13/18 (72%) showed persistent abnormalities (e.g., reduced LVEF and/or coronary changes).

One patient (eight-year-old, female, non-Kuwaiti, with no co-morbid conditions) demised due to cardiogenic shock. This child had myocardial dysfunction and pericarditis on echo findings. In addition, the child had high troponin T, D-dimer, and ferritin levels (>500 ngr/mL). The child was admitted to the PICU for six days prior to her demise.

## 4. Discussion

The incidence and admission patterns in Kuwait reflect the global trend in SARS-CoV-2 infections. The highest proportion of patients was admitted during the initial alpha strain of SARS-CoV-2 between February 2020 and November 2020. There was a decline in incidence during the Beta variant, followed by a subsequent rise in incidence, with 39% of cases occurring during the Delta variant.

Globally, the proportion of MIS-C cases was highest during the initial SARS-CoV-2 wave and peaked during the fourth wave (Delta variant strain) from May to December 2021 [22], as evident from a study at the University Children’s Hospital of Cracow, which showed that 74 (68.5%) out of 108 patients suspected of MIS-C were hospitalized between 1 November 2020 and 30 June 2022, and 34 patients (31.5%) were hospitalized when the Delta variant was dominant (October 2021–June 2022) [23]. However, data from the United Kingdom showed that MIS-C was less common during the Delta and Omicron variants. Findings from a study in Cape Town also showed that the numbers of MIS-C admissions for the first and fourth waves were similar [24].

The age distribution was skewed toward younger children, with preschool and early-school ages predominating (see Table 2 (a)). The age distribution in our study shares similarity with findings from a cohort study in Saudi Arabia, in which 30% of children with MIS-C were under 2 years old and 30% were over 5 years [25]. However, the cohort in our study is younger than those described in several systematic reviews and meta-analyses that indicate a mean age of children with MIS-C in the region of 8 to 9 years [26,27], whilst other reviews have shown a median age across included studies between 7 to 10 years, with an age range spanning from 7 months to 20 years [27]. The similarity in age distribution with Saudi Arabia could potentially be due to a sharing of genetic make-up [28]. The lower age profile observed in this cohort is more plausibly explained by demographic composition, the timing of epidemic waves/variants, and local testing/admission thresholds than by a distinct local etiology. Because the registry records nationality as residency (not ancestry) and did not capture ethnic/genetic data, causal inferences about etiology cannot be made (2).

Consistent with findings from the systematic reviews and meta-analyses, our study has a slight male predominance and has shown a lower level (22% vs. 35%) of co-morbidities in reported cases [27,28,29]. The clinical presentation of the patients in the current cohort showed that gastrointestinal symptoms, bilateral conjunctivitis, and skin rashes were the most common presenting symptoms, while respiratory symptoms were less frequent. Similar presenting symptoms were reported from a Children’s Medical Centre Hospital in Tehran, Iran [29], as well as in other systematic reviews, which indicated fever, gastrointestinal symptoms, rash, and bilateral conjunctivitis as the predominant symptoms [30,31].

Clinical outcomes in terms of mortality (1%), intensive care unit admission (42%), median duration of ICU stay of four days (IQR: 2.5–6.5), prescription of antimicrobial treatment, intravenous immunoglobulins, and corticosteroids are in keeping with the published studies [27]. However, the proportion of patients admitted to the intensive care unit in the current cohort was higher than those reported in Iran (21%) [29].

In the current cohort, a lesser proportion of patients than reported in a meta-analysis required respiratory support, with only 17% being ventilated in comparison to findings reported from four tertiary care centers in Paris that showed 40% of the patients required mechanical ventilation [32]. In addition, a study conducted in the United States showed that 43% received supplemental oxygen and 25% required non-invasive ventilation, while acute kidney injury was seen in 21% of patients [33]. These findings were based on the original Wuhan phase of the pandemic.

Previous studies, including a systematic review, have noted that cardiac involvement occurs in up to 67–80% of children with MIS-C [34,35]. Cardiac manifestations can vary in severity from mild to severe or critical, in the form of cardiogenic shock [36]. Cardiac signs include ventricular dysfunction, coronary artery aneurysms, conduction abnormalities, and arrhythmias [37].

Nineteen percent (16/83) of the patients with MIS-C developed cardiac abnormalities. Patients with severe SARS-CoV-2 were significantly more likely to have cardiac abnormalities. Coronary abnormalities and myocardial dysfunction were the most common abnormalities detected. Two patients with dilated coronary arteries had aneurysms (one classified as small and the other as medium size). The pathophysiological mechanism for the coronary artery dilatation is attributed to the cytokine storm, similar to the one experienced by patients with juvenile idiopathic arthritis [38]. Myocardial dysfunction is postulated to be a result of direct myocardial damage caused by the virus entering cardiomyocytes via the angiotensin-converting enzyme 2 (ACE2) receptor, as well as cell-mediated cytotoxicity through the myocardial inflammation resulting from CD8+ T lymphocyte migration to cardiomyocytes [39].

The mortality rate of 1% is like other higher income countries [40]. The clinical profile of septic shock, high levels of ferritin, D-dimer, and troponin T, as well as myocardial dysfunction and pericarditis, is reflective of the markers for an increased risk of mortality [41].

Cardiac involvement in MIS-C is unavoidable [42]. In our study, those with follow-up echocardiography at approximately 3 months (*n* = 18), 13/18 (72%) showed persistent abnormalities. A total of 32.5% (*n* = 27/83) of children had complications ranging from myocardial dysfunction to dilated coronary arteries and pericardial effusions three months following their discharge from hospital. There are limited studies that have established the long-term consequences of cardiac involvement in MIS-C in children.

In exploratory analyses, higher troponin tended to track with lower LVEF, whereas CRP, ferritin, and D-dimer showed no consistent relationships with echocardiographic abnormalities. None of these associations reached statistical significance in this cohort, suggesting that single-timepoint biomarkers alone may have limited discriminatory value for cardiac involvement in MIS-C.

### Strengths and Limitations

This study is the first to be generated within the Gulf region using a population-based registry to describe the epidemiological and clinical features of children less than 14 years of age with SARS-CoV-2-related MIS-C. The findings highlight similarities and a few unique characteristics applicable to children residing in Kuwait.

The limitations of this study include its retrospective and descriptive nature. This study has several limitations. First, its retrospective, registry-based, and descriptive design precludes causal inference and limits control of confounding. Second, although the case adjudication used WHO criteria, borderline presentations may have been misclassified as MIS-C or non-MIS-C; provisional cases that were not confirmed were not systematically followed, so delayed reclassification cannot be excluded. Third, missing data, including the inconsistent capture of quantitative LVEF (numeric values not recorded at some sites) and incomplete laboratory/imaging panels, may bias estimates and reduce precision. The sample size limits power for outcomes and multivariable adjustment. The limited sample size and missing data may have obscured small but clinically relevant effects. Clinical management was not standardized (e.g., timing of IVIG/steroids, PICU thresholds), which may influence the observed outcome.

## 5. Conclusions

This study shows a younger age of patients being admitted with MIS-C than other global comparisons, and shows more local testing/admission pathways than inherent biological differences, particularly as the registry records nationality as residency rather than ancestry. Cardiac involvement centered on coronary changes and left-ventricular dysfunction, with lower LVEF among those with echocardiographic abnormalities. Critical-care utilization was substantial, whereas invasive mechanical ventilation was minimal.

These findings support a standardized cardiac pathway for suspected MIS-C—early ECG and echocardiography for all, serial inflammatory/cardiac biomarkers for risk-stratification, timely cardiology inputs, and structured follow-ups (e.g., clinic review and repeat echocardiography at 4–6 weeks, and again around 3 months for abnormalities). Service planning should prioritize PICU capacity for haemodynamic support and infusion therapies (IVIG/steroids) over ventilator demand and ensure equitable an access for non-Kuwaiti residents. For future improvement, the routine capture of numeric LVEF, assay reference intervals, and longitudinal outcomes in registries will enable benchmarking and prognostic modeling; prospective analytic studies are warranted to identify predictors of PICU admission, cardiac involvement, and recovery, and to evaluate the timing and effectiveness of immunomodulatory regimens.

## Figures and Tables

**Figure 1 diagnostics-15-02545-f001:**
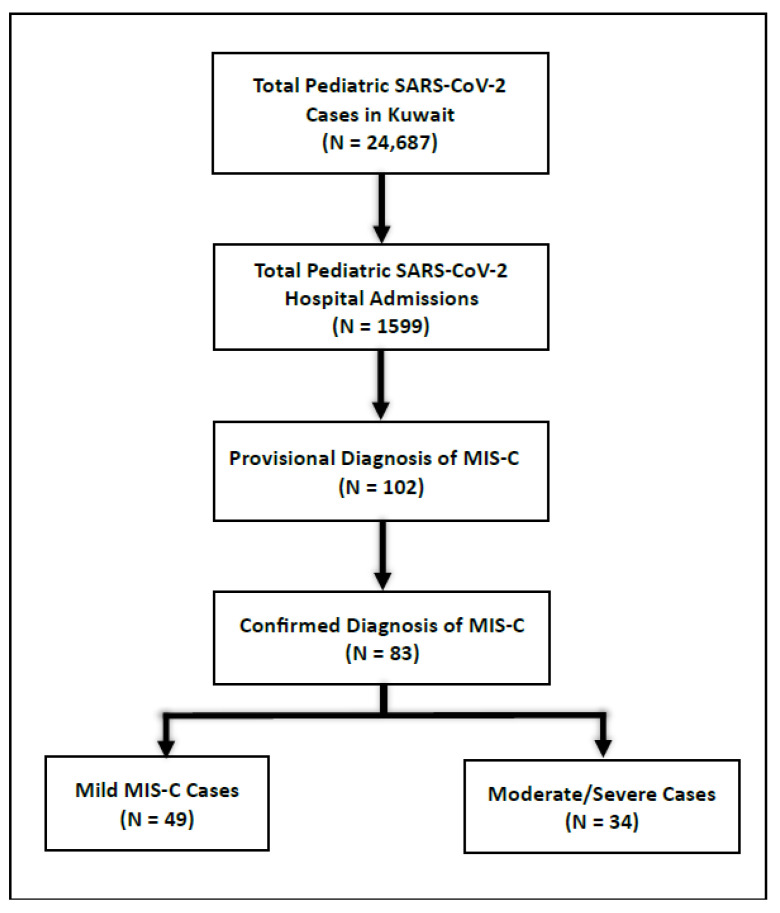
Study population breakdown (February 2020–February 2022).

**Figure 2 diagnostics-15-02545-f002:**
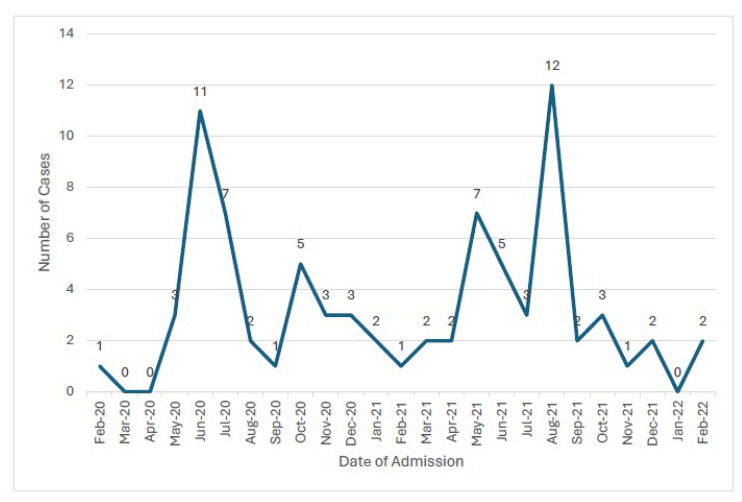
Hospital admission of MIS-C by month (February 2020–February 2022).

**Figure 3 diagnostics-15-02545-f003:**
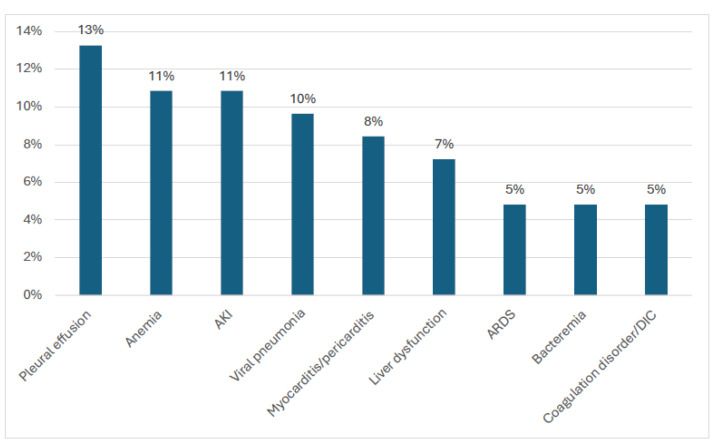
Common complications detected.

**Figure 4 diagnostics-15-02545-f004:**
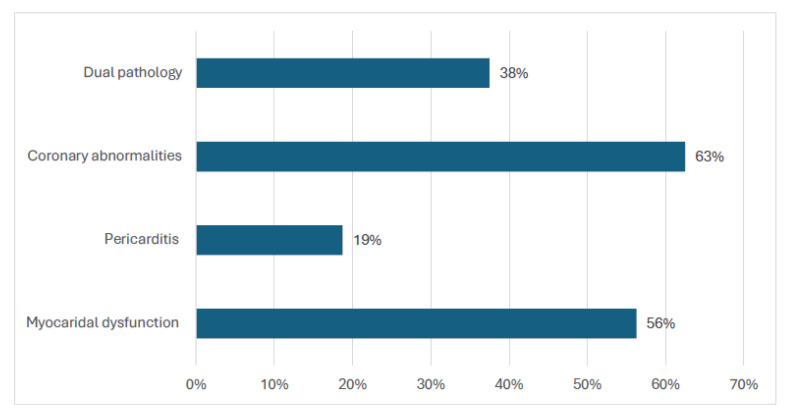
Cardiac abnormalities diagnosed amongst children with MIS-C in Kuwait 2020–2021.

**Table 1 diagnostics-15-02545-t001:** Preliminary case definition of MIS-C (15 May 2020) [18].

Children and adolescents 0–19 years of age with fever ≥3 days
AND two of the following: Rash or bilateral non-purulent conjunctivitis or muco-cutaneous inflammation signs (oral, hands or feet). Hypotension or shock. Features of myocardial dysfunction, pericarditis, valvulitis, or coronary abnormalities (including ECHO findings or elevated Troponin/NT-pro BNP). Evidence of coagulopathy (by PT, PTT, elevated D-dimers). Acute gastrointestinal problems (diarrhea, vomiting, or abdominal pain).
AND Elevated markers of inflammation such as ESR, C-reactive protein, or procalcitonin.
AND No other obvious microbial cause of inflammation, including bacterial sepsis, staphylococcal or streptococcal shock syndromes.
AND Evidence of COVID-19 (RT-PCR, antigen test or serology positive) or likely contact with patients with COVID-19.

**Table 2 diagnostics-15-02545-t002:** (**a**) Demographic characteristics of children with MIS-C. (**b**) Laboratory parameters of children with MIS-C. * *p* < 0.05 indicates statistical significance.

(a)						
Variable	Characteristic	No Cardiac Abnormality	Cardiac Abnormality	Total (84)	Chi Squared	*p* Value
Gender	Male	26 (43%)	8 (10%)	44 (52%)	0.58	0.45
	Female	30 (36%)	10 (12%)	40 (48%)		
Nationality	Kuwait	36 (43%)	5 (6%)	41 (49%)	4.06	0.04
	Non-Kuwaiti	30 (36%)	13 (15%)	43 (51%)		
Age group	0–3 years	23 (27%)	8 (10%)	31 (37%)	4.5	0.21
	4 years–6 years	20 (24%)	3 (4%)	23 (27%)		
	7 years–9 years	15 (18%)	2 (2%)	17 (20%)		
	>9 years	8 (12%)	5 (6%)	13 (16%)		
Disease severity	Mild	46 (55%)	7 (8%)	53 (63%)	7.61	0.02 *
	Moderate	15 (18%)	6 (7%)	21 (25%)		
	Severe	5 (6%)	5 (6%)	10 (12%)		
**(b)**						
**Laboratory Variables**		**MIS-C (Mean/SD)**	**Median (IQR)**	**Normal Values**		
Ferritin (μg/L)		637 (790)	414 (133–819)	7–140		
C-reactive protein (mg/L)		175 (114)	157 (92–237)	0–8		
Troponin T (ng/L)		44 (104)	10 (1.5–31)	<14		
D-dimer (μg/L)		1953 (2103)	1289 (803–1800)	<250		

**Table 3 diagnostics-15-02545-t003:** Demographic profile and laboratory parameters and cardiac abnormalities.

Variable	Level	No Cardiac Abnormality *n* (%)	Cardiac Abnormality *n* (%)	Total *n* (%)	*p*-Value
Gender	0	30 (36.6%)	9 (11.0%)	39 (47.6%)	0.578 (Fisher)
	1	36 (43.9%)	7 (8.5%)	43 (52.4%)	
Nationality	0	36 (43.9%)	7 (8.5%)	43 (52.4%)	0.578 (Fisher)
	1	30 (36.6%)	9 (11.0%)	39 (47.6%)	
Age group	0	14 (17.1%)	3 (3.7%)	17 (20.7%)	0.630 (χ^2^)
	1	25 (30.5%)	7 (8.5%)	32 (39.0%)	
	2	21 (25.6%)	6 (7.3%)	27 (32.9%)	
	3	6 (7.3%)	0 (0.0%)	6 (7.3%)	
Disease severity	Better	2 (2.4%)	0 (0.0%)	2 (2.4%)	0.630 (χ^2^)
	Mild	36 (43.9%)	10 (12.2%)	46 (56.1%)	
	Moderate	17 (20.7%)	5 (6.1%)	22 (26.8%)	
	Severe	11 (13.4%)	1 (1.2%)	12 (14.6%)	
**Measure**	**No Cardiac** **Abnormality** **Mean (SD)**		**Cardiac** **Abnormality** **Mean (SD)**	**Total** **Mean (SD)**	***p*-Value**
Ferritin (µg/L)	800.06 (1778.39)		594.17 (952.58)	756.71 (1634.13)	0.277
C-reactive protein (mg/L)	143.64 (107.27)		192.64 (136.05)	153.44 (114.30)	0.236
Troponin (ng/L)	37.87 (99.30)		24.06 (33.71)	35.30 (90.64)	0.216
D-dimer (µg/L FEU)	1608.06 (1586.02)		2120.43 (2698.43)	1715.12 (1860.15)	0.982

**Table 4 diagnostics-15-02545-t004:** Cardiac biomarkers and echocardiographic findings in MIS-C (available-case): Panel A—biomarkers by echo status; Panel B—LVEF–biomarker correlations; Panel C—LVEF by echo status.

Marker	Median (IQR) Normal	Median (IQR) Abnormal	Exact *p*
Troponin	4.96 (0.45–22.84)	12.00 (7.37–21.40)	0.2161
D-dimer	1193.00 (753.00–1665.00)	1087.00 (765.25–1817.25)	0.9815
Ferritin	413.60 (167.00–697.00)	192.75 (92.75–850.25)	0.2773
CRP	138.00 (66.60–202.68)	172.00 (87.20–286.00)	0.2358
**Comparison LVEF vs. biomarkers**	**Spearman ρ**	**Exact *p***	
LVEF vs. Troponin	−0.199	0.3208	
LVEF vs. D-dimer	0.183	0.3245	
LVEF vs. Ferritin	0.111	0.5732	
LVEF vs. CRP	−0.271	0.1045	
Median (IQR) abnormal (LVEF vs. Echocardiogram)	**Exact *p***		
56.00 (50.50–65.00)	0.1003		

## Data Availability

The data that support the findings of this study are available from Ministry of Health in Kuwait, but restrictions apply to the availability of these data, which were used under license for the current study, and so are not publicly available. Data are, however, available from the authors upon reasonable request and with permission of the Ministry of Health.

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
