# Peer review of "Incidence, Clinical Profile, and Cardiac Manifestations of MIS-C in Children in Kuwait"

_diagnostics, 2025, doi:10.3390/diagnostics15192545_

Round 1

Reviewer 1 Report

Comments and Suggestions for Authors

               MIS-C remains a serious post-COVID syndrome in children, and studying its incidence and features in diverse settings is important. The topic is timely and clinically significant. Focusing on Kuwait fills an evidence gap, given limited MIS-C data from the Middle East.

  1. The list of collected variables is comprehensive. For reproducibility, it would help to state whether all children meeting MIS-C criteria received echocardiography and ECG by protocol. Also clarify how missing data were handled (e.g. were analyses done on available cases, were any imputations performed?).
  2. Methods – Statistical Analysis (Lines 481–489): The description of data analysis is very brief. Please specify which statistical tests were used (e.g. chi-square for proportions, t-test or Mann-Whitney for means). The text notes a significant LVEF difference (p<0.05) but does not say how this was tested. Please indicate the test and exact p-value. Also, “Statistical Software for Data Science version 18” is unclear – name the software (e.g. SPSS, Stata) and version explicitly.
  3. Results – Consider clarifying in text that the incidence of MIS-C is 83 out of 24,637 infections (≈0.34%, or 3.32 per 1000 infections). Also, explain the drop from 102 provisional to 83 final cases (19 did not meet criteria); were these followed, and could misclassification be a concern (as noted in limitations)?
  4. Results – Demographics and Clinical Features (Lines 576–585; Table 2): The age distribution and gender/nationality split are described. It is interesting that over half were non-Kuwaiti. The table indicates laboratory markers (ferritin, CRP, troponin, D-dimer) with means and medians. Please ensure units and normal ranges are either in the table or text for context. For example, an elevated troponin median of 10 ng/L may be hard to interpret without a reference. Also, the table format is somewhat confusing as it mixes categories and lab values. Consider splitting demographics (age, sex, severity) into one table and labs into another, to improve readability.
  5. Results – Cardiac Findings (Lines 704–713): The identification of 16 patients with cardiac abnormalities is clear. The LVEF finding is noted (mean 61.25%; lower in the cardiac group, p<0.05) – specify how many patients had LVEF recorded. If only 50% had echo data, note whether the others had normal echos or missing data.
  6. Results – Treatment and Outcomes (Lines 756–764; 771–779): The interventions and outcomes are reported. One discrepancy: the text here says 47% required ICU (39/83), but the Discussion later says 42% ICU (likely an error). Please correct to match. Likewise, mechanical ventilation is given as 6% (5/83), whereas the Discussion states 17% of patients were ventilated (17% would correspond to ~14/83, so it seems the Discussion figure is off). Check all percentage calculations against the raw counts. In “residual sequelae,” it would be good to specify if the 18 patients with LVEF <55% were those not fully recovered at discharge, or if follow-up was done. The note of one death is important – the detailed description is appropriate.
  7. Discussion – Alignment with data: Some discussion statements should be checked against results for consistency: as noted, ICU% and ventilation%. Also, the sentence “Cardiac involvement in MIS-C is unavoidable [44]. In our study, 13 (72%) children had complications…” seems to refer to a subset at 3 months follow-up (“three months following their discharge”). This outcome (72% with ongoing complications) was not described in the Results section, and it’s unclear if this is from the current study or another reference. Clarify whether there was any longitudinal follow-up for the cohort. If not, remove or rephrase to avoid confusion.
  8. The abbreviation list should spell “Electrocardiogram” correctly (it is cut “Electrocardiogran”).

Overall, this paper provides a useful contribution on MIS-C in a population that has been under-reported. With clarification of the few inconsistencies noted above and minor reformatting of tables/figures for clarity, it can be a valuable reference for pediatric COVID research. The authors’ detailed methodology and transparent reporting will aid reproducibility. Addressing the points above will strengthen the manuscript and its interpretability.

Author Response

MIS-C remains a serious post-COVID syndrome in children, and studying its incidence and features in diverse settings is important. The topic is timely and clinically significant. Focusing on Kuwait fills an evidence gap, given limited MIS-C data from the Middle East.

Comment 1: The list of collected variables is comprehensive. For reproducibility, it would help to state whether all children meeting MIS-C criteria received echocardiography and ECG by protocol. Also clarify how missing data were handled (e.g. were analyses done on available cases, were any imputations performed?).

Response: 

all children meeting MIS-C criteria received echocardiography and ECG by protocol added to line 144

Line 146-150: Analyses were conducted on available cases, and no single or multiple imputation was performed. For each variable, the analysis denominator reflects the number of non-missing observations and is reported in the text/tables as n/N. Variables captured by protocol for all MIS-C cases (case status, MIS-C severity, PICU admission, echocardiography and ECG) were complete. No values were substituted or carried forward

Comment 2: Statistical Analysis (Lines 481–489): The description of data analysis is very brief. Please specify which statistical tests were used (e.g. chi-square for proportions, t-test or Mann-Whitney for means). The text notes a significant LVEF difference (p<0.05) but does not say how this was tested. Please indicate the test and exact p-value. Also, “Statistical Software for Data Science version 18” is unclear – name the software (e.g. SPSS, Stata) and version explicitly.

Response: Data were analysed in Stata 18 (StataCorp LLC, College Station, TX, USA) (Line 138). Categorical variables are presented as counts and percentages; continuous variables as mean (SD) if approximately normal or median (IQR) otherwise. sts. Between-group comparisons (2 groups): Pearson’s chi-square or Fisher’s exact test (categorical); Across MIS-C severity (3 groups): Chi-square/Fisher’s exact (categorical); Incidence proportions: reported with 95% confidence intervals

Comment 3: Consider clarifying in text that the incidence of MIS-C is 83 out of 24,637 infections (≈0.34%, or 3.32 per 1000 infections). Also, explain the drop from 102 provisional to 83 final cases (19 did not meet criteria); were these followed, and could misclassification be a concern (as noted in limitations)?

Response: Line 1179-193 modified as follows: Between February 2020 and November 2021, 24,637 children with SARS-CoV-2 were recorded in PCR-Q8; 1,599/24,637 (6.5%) were hospitalised. Of these admissions, 102/1,599 (6.4%) received a provisional diagnosis of MIS-C and 83/1,599 (5.2%) were confirmed after multidisciplinary review. The incidence proportion of MIS-C among infections was 83/24,637 = 0.34%, i.e., 3.37 per 1,000 infections (95% CI ≈ 2.7–4.2 per 1,000), and 5.19 per 100 among hospital admissions (83/1,599). Most confirmed MIS-C cases were mild (49/83, 59%), followed by moderate (22/83, 27%) and severe (12/83, 14%). (Figure 1). Among 102 provisional MIS-C cases, 19 (18.6%) were not confirmed on adjudication because full WHO criteria were not met and/or an alternative diagnosis was favoured; these children were retained in the overall SARS-CoV-2 cohort but excluded from MIS-C analyses

Comment 4: Demographics and Clinical Features (Lines 576–585; Table 2): The age distribution and gender/nationality split are described. It is interesting that over half were non-Kuwaiti. The table indicates laboratory markers (ferritin, CRP, troponin, D-dimer) with means and medians. Please ensure units and normal ranges are either in the table or text for context. For example, an elevated troponin median of 10 ng/L may be hard to interpret without a reference. Also, the table format is somewhat confusing as it mixes categories and lab values. Consider splitting demographics (age, sex, severity) into one table and labs into another, to improve readability.

Response: Adjusted in line 210. 

Comment 5: Cardiac Findings (Lines 704–713): The identification of 16 patients with cardiac abnormalities is clear. The LVEF finding is noted (mean 61.25%; lower in the cardiac group, p<0.05) – specify how many patients had LVEF recorded. If only 50% had echo data, note whether the others had normal echos or missing data

Response: Line 25-251: There was no numeric LVEF captured in the registry

Comment 6: Treatment and Outcomes (Lines 756–764; 771–779): The interventions and outcomes are reported. One discrepancy: the text here says 47% required ICU (39/83), but the Discussion later says 42% ICU (likely an error). Please correct to match. Likewise, mechanical ventilation is given as 6% (5/83), whereas the Discussion states 17% of patients were ventilated (17% would correspond to ~14/83, so it seems the Discussion figure is off). Check all percentage calculations against the raw counts. In “residual sequelae,” it would be good to specify if the 18 patients with LVEF <55% were those not fully recovered at discharge, or if follow-up was done. The note of one death is important – the detailed description is appropriate.

Response: We corrected inconsistencies: PICU/ICU admission is 39/83 (47%) throughout, and invasive mechanical ventilation is 5/83 (6%). We now specify that 27/83 (33%) had residual cardiac sequelae at discharge, of whom 18/83 (22%) had LVEF <55%. We also report 3-month follow-up echocardiography where available.

Comment 7: Some discussion statements should be checked against results for consistency: as noted, ICU% and ventilation%. Also, the sentence “Cardiac involvement in MIS-C is unavoidable [44]. In our study, 13 (72%) children had complications…” seems to refer to a subset at 3 months follow-up (“three months following their discharge”). This outcome (72% with ongoing complications) was not described in the Results section, and it’s unclear if this is from the current study or another reference. Clarify whether there was any longitudinal follow-up for the cohort. If not, remove or rephrase to avoid confusion. 

Response: We corrected the Discussion to match the Results (PICU admission 39/83, 47%; invasive ventilation 5/83, 6%). Regarding the “72% at 3 months” statement. We added the corresponding sentence and denominator to the Results (Section 3.4) and now reference it explicitly in the Discussion.

Comment 8: The abbreviation list should spell “Electrocardiogram” correctly (it is cut “Electrocardiogran”).

Response: Correction made in abbreviation list 

Reviewer 2 Report

Comments and Suggestions for Authors

Thank you for having the opportunity to read the article entitled "Incidence, Clinical Profile and Cardiac Manifestations of MIS-C in Children in Kuwait". It is interesting, but I would have some suggestions for the authors:

1. Don't repret the results from the table in the text (e.g. a quarter (25%) etc.). In the manuscript, the results just need to be interpreted, not repeated.

2. The authors mentioned that "MIS-C appears to affect younger children in Kuwait than in other countries" but then they mentioned that 52% were non-Kuwaiti. Could this influence the evolution, regarding for example a local ethiology?

3. This is just a descriptive study. Therefore, more relevant descriptive parameter should be introduced - e.g. heredocollateral antecedents, treatment etc. Despite the fact that is a descriptive study, some association between clinical/paraclinical evolution and diagnostics are needed to be performed. The statistical analysis was very briefly made.

4. There are more limitations of the study.

5. In the discussion section, the authors should provide also some explanation and clinical implication regarding the results.

Author Response

Comment 1: Don't repeat the results from the table in the text (e.g. a quarter (25%) etc.). In the manuscript, the results just need to be interpreted, not repeated

Response: We have revised the Discussion to remove numeric percentages and retain interpretation only; detailed counts remain in Table 2a.

Comment 2: The authors mentioned that "MIS-C appears to affect younger children in Kuwait than in other countries" but then they mentioned that 52% were non-Kuwaiti. Could this influence the evolution, regarding for example a local ethiology?

Response: The lower age profile observed in this cohort is more plausibly explained by demographic composition, timing of epidemic waves/variants, and local testing/admission thresholds than by a distinct local etiology. Because the registry records nationality as residency (not ancestry), and did not capture ethnic/genetic data, causal inferences about etiology cannot be made

Comment 3: This is just a descriptive study. Therefore, more relevant descriptive parameter should be introduced - e.g. heredocollateral antecedents, treatment etc. Despite the fact that is a descriptive study, some association between clinical/paraclinical evolution and diagnostics are needed to be performed. The statistical analysis was very briefly made.

Response: We agree that this was a purely descriptive study. Family history and certain longitudinal outcomes were incompletely captured in the registry, limiting inference about predisposition and evolution over time.

Comment 4: There are more limitations of the study

Response: This study has several limitations. First, its retrospective, registry-based and descriptive design precludes causal inference and limits control of confounding. Second, although case adjudication used WHO criteria, borderline presentations may have been misclassified as MIS-C or non–MIS-C; provisional cases that were not confirmed were not systematically followed, so delayed reclassification cannot be excluded. Third, missing data, including inconsistent capture of quantitative LVEF (numeric values not recorded at some sites) and incomplete laboratory/imaging panels, required an available-case approach without imputation, which may bias estimates and reduces precision; we mitigate this by reporting n/N denominators for all summaries. Fourth, data capture varied by site and over time (e.g., assay platforms and reference intervals for troponin, D-dimer, ferritin; evolving clinical protocols across waves), introducing measurement and temporal bias. Fifth, the registry lacked key covariates (e.g., ethnicity/genetic ancestry, socioeconomic indicators, standardized family history, and complete longitudinal outcomes), limiting assessment of predisposition and evolution over time. Sixth, findings reflect hospitalized children <14 years in Kuwait’s public system; mild community cases, private-sector encounters, and older adolescents are under-represented, which constrains external generalizability. Seventh, the sample size limits power for uncommon outcomes and multivariable adjustment; our association analyses are therefore exploratory/hypothesis-generating only. Finally, clinical management was not standardized (e.g., timing of IVIG/steroids, PICU thresholds), which may influence observed outcomes.

Comment 5: In the discussion section, the authors should provide also some explanation and clinical implication regarding the results.

Response: Line 401-420: This pattern is more plausibly explained by population structure, wave/variant timing, and local testing/admission pathways than by inherent biological differences, particularly as the registry records nationality as residency rather than ancestry. Cardiac involvement centered on coronary changes and left-ventricular dysfunction, with lower LVEF among those with echocardiographic abnormalities. Critical-care utilization was substantial, whereas invasive mechanical ventilation was minimal. These findings support a standardized cardiac pathway for suspected MIS-C—early ECG and echocardiography for all, serial inflammatory/cardiac biomarkers for risk-stratification, timely cardiology input, and structured follow-up (e.g., clinic review and repeat echocardiography at 4–6 weeks and again around three months for abnormalities). Service planning should prioritize PICU capacity for hemodynamic support and infusion therapies (IVIG/steroids) over ventilator demand, and ensure equitable access for non-Kuwaiti residents. For future improvement, routine capture of numeric LVEF, assay reference intervals, and longitudinal outcomes in registries will enable benchmarking and prognostic modelling; prospective analytic studies are warranted to identify predictors of PICU admission, cardiac involvement, and recovery, and to evaluate timing and effectiveness of immunomodulatory regimens.

Reviewer 3 Report

Comments and Suggestions for Authors

25-previous non-acute infection or specify whether we are talking about MISC-C or PIMS-TS patients
48-acute COVID or MIS-C?
103/ tab 1-these are the criteria for PIMS-TS (as  bibliographic ref 18), the criteria chosen to define MIS-C from among those of the WHO, CDC, etc. must be specified
172-173 specify which tests the patients underwent in addition to IgM testing
204- specify what type of coronary anomalies we are talking about: ectasia? aneurysms? of what calibre? so that they can also be correlated with their regression/permanence (line 221)
222-specify the follow-up interval. Three months, as indicated in lines 306-308?
252-254- when comparing the data with those provided in Table 2, the numbers do not match (840% not 37% and 20% not 25%)
289-here too, there is no clarity between acute COVID and MIS-C
294-I would rather mention Kawasaki syndrome and the differential diagnosis in the bibliography
306-with reference to line 220, weren't there 27 patients?

General comment:
I think it should be clarified whether the population is affected by COVID in the acute phase, whether we are talking about MIS-C or paediatric inflammatory multisystem syndrome temporally associated with COVID-19 (PIMS-TS) (1). It should therefore be clarified whether patients underwent a blood test or a broncho-alveolar lavage  rather than a nasopharyngeal swab to test for positivity. Furthermore, IgG positivity is not specified (2). This would suggest acute infection in a number of patients, a hypothesis supported by the fact that the percentage of MIS-C patients under the age of 6 (59%) is significantly higher than reported in multicentre studies (3)

1-Harwood R, Allin B, Jones CE, Whittaker E, Ramnarayan P, Ramanan AV, Kaleem M, Tulloh R, Peters MJ, Almond S, Davis PJ, Levin M, Tometzki A, Faust SN, Knight M, Kenny S; PIMS-TS National Consensus Management Study Group. A national consensus management pathway for paediatric inflammatory multisystem syndrome temporally associated with COVID-19 (PIMS-TS): results of a national Delphi process. Lancet Child Adolesc Health. 2021 Feb;5(2):133-141. doi: 10.1016/S2352-4642(20)30304-7. Epub 2020 Sep 18. Erratum in: Lancet Child Adolesc Health. 2021 Feb;5(2):e5. doi: 10.1016/S2352-4642(20)30400-4. PMID: 32956615; PMCID: PMC7500943.
2-Ghazizadeh Esslami G, Mamishi S, Pourakbari B, Mahmoudi S. Systematic review and meta-analysis on the serological, immunological, and cardiac parameters of the multisystem inflammatory syndrome (MIS-C) associated with SARS-CoV-2 infection. J Med Virol. 2023 Jul;95(7):e28927. doi: 10.1002/jmv.28927. PMID: 37436781.
3-Algarni AS, Alamri NM, Khayat NZ, Alabdali RA, Alsubhi RS, Alghamdi SH. Clinical practice guidelines in multisystem inflammatory syndrome (MIS-C) related to COVID-19: a critical review and recommendations. World J Pediatr. 2022 Feb;18(2):83-90. doi: 10.1007/s12519-021-00499-w. Epub 2022 Jan 4. PMID: 34982402; PMCID: PMC8725428.

Author Response

Comment 1: 25-previous non-acute infection or specify whether we are talking about MISC-C or PIMS-TS patients. 

Response: We revised the abstract to post-acute hyperinflammatory condition that occurs in children 2–6 weeks after Severe Acute Respiratory Syndrome Coronavirus 2 (SARS-CoV-2), infection or exposure), varies between countries. We also state that we used the WHO MIS-C case definition (equivalent to PIMS-TS per RCPCH) and use the term MIS-C throughout. 

Comment 2: Line 48- acute COVID or MIS-C?

Response: we have clarified that our cohort comprises post-acute MIS-C, not acute COVID-19. Case ascertainment followed the WHO preliminary MIS-C definition (15 May 2020) (equivalent to PIMS-TS per RCPCH). 

Comment 3: 103/ tab 1-these are the criteria for PIMS-TS (as  bibliographic ref 18), the criteria chosen to define MIS-C from among those of the WHO, CDC, etc. must be specified

Response: Kuwait’s Pediatric COVID-19 Task Force adopted the WHO preliminary MIS-C definition (15 May 2020) as the operational standard for case confirmation and national surveillance (Table 1). The RCPCH PIMS-TS (UK) describes a similar pediatric hyper-inflammatory syndrome but does not mandate laboratory/epidemiologic linkage to SARS-CoV-2; CDC MIS-C is closely aligned with WHO but differs modestly in age and fever thresholds. In our registry, children who met PIMS-TS features without a WHO-compatible SARS-CoV-2 link were not classified as MIS-C. 

Comment 4: 172-173 specify which tests the patients underwent in addition to IgM testing

Response: This was an error- and should reflect IgG. 

Comment 5: 204- specify what type of coronary anomalies we are talking about: ectasia? aneurysms? of what calibre? so that they can also be correlated with their regression/permanence (line 221)

Response: We modified the sentence as follows: Sixty three percent % (10/16) of the patients had cardiac involvement. Of these nine patients (56%) had myocardial dysfunction, whilst six patients (38%) had dual pathologies. 

Comment 6: 222- specify the follow-up interval. Three months, as indicated in lines 306-308

Response: Three months added

Comment 7: 252-254- when comparing the data with those provided in Table 2, the numbers do not match (840% not 37% and 20% not 25%)

Response: Table 2 as been fixed

Comment 8: 289-here too, there is no clarity between acute COVID and MIS-C

Response: Clarified as MISC

Comment 9: I would rather mention Kawasaki syndrome and the differential diagnosis in the bibliography

Response: 

Comment 10: 306-with reference to line 220, weren't there 27 patients?

Response: Among those with follow-up echocardiography at approximately 3 months (n=18), 13/18 (72%) showed persistent abnormalities (e.g., reduced LVEF and/or coronary changes).

Round 2

Reviewer 1 Report

Comments and Suggestions for Authors

All my comments have been adequately addressed. Please recheck the reference list.

Author Response

Comment: All my comments have been adequately addressed. Please recheck the reference list.

Response: Thank you for pointing this out.

Reviewer 2 Report

Comments and Suggestions for Authors

Thank you for making the changes that I suggested. However, an important aspect for the statistic analysis for an original paper is to make some correlation between the results, as I previously suggested. For example, between gender and laboratory findings, between laboratory findings (especially troponin) and  echocardiographic parameters etc.

Author Response

Comment: Thank you for making the changes that I suggested. However, an important aspect for the statistic analysis for an original paper is to make some correlation between the results, as I previously suggested. For example, between gender and laboratory findings, between laboratory findings (especially troponin) and  echocardiographic parameters etc.

Response: 

Adjusted: Lines 274-284 - Compared with children without echocardiographic abnormalities, those with any abnormality showed a non-statistically significant higher troponin concentration (median 12.00 [IQR 7.37–21.40] vs 4.96 [0.45–22.84] ng/L), (Mann–Whitney U p=0.2161). D-dimer, ferritin, and CRP were similar between groups (Table 4). LVEF was lower among children with any echocardiographic abnormality (median 56.0% [50.5–65.0]), although the difference versus those without abnormality was not significant (Mann–Whitney U p=0.1003). In correlation analyses, troponin was inversely related to LVEF (Spearman ρ=−0.199, p=0.3208), while D-dimer (ρ=0.183, p=0.3245) and ferritin (ρ=0.111, p=0.5732) showed weak positive, non-significant correlations, and CRP trended inversely (ρ=−0.271, p=0.1045). Overall, there was no biomarker–echo associations in this cohort, with estimates limited by sample size and missing values.